**Data Availability Statement:** Data cannot be shared publicly because it contains personal data

# Risk of dementia and Parkinson's disease in patients treated with androgen deprivation therapy using gonadotropin-releasing hormone agonist for prostate cancer: A nationwide population-based cohort study

**Myungsun Shim[1], Woo Jin Bang[1], Cheol Young Oh[1], Yong Seong Lee[1], Seong Soo Jeon[2], Hanjong Ahn[3], Young-Su Ju[4], Jin Seon Cho●[1] \***

**1** Department of Urology, Hallym University College of Medicine, Hallym University Sacred Heart Hospital, Anyang, GyeongGi-Do, Korea, **2** Department of Urology, Sungkyunkwan University School of Medicine, Samsung Medical Center, Seoul, Korea, **3** Department of Urology, University of Ulsan College of Medicine, Asan Medical Center, Seoul, Korea, **4** Department of Occupational and Environmental Medicine, Hallym University College of Medicine, Anyang, GyeongGi-Do, Korea

\* js315@hallym.or.kr

## Abstract

Recent studies reported conflicting results on the association of androgen deprivation therapy (ADT) with dementia and Parkinson's disease in patients with prostate cancer (Pca). Therefore, this study aimed to investigate whether use of gonadotropin-releasing hormone agonist (GnRHa) increases the risk of both diseases. A nationwide population cohort study was conducted involving newly diagnosed patients with Pca %who started ADT with GnRHa (GnRHa users, n = 3,201) and control (nonusers, n = 4,123) between January 1, 2012, and December 31, 2016, using data from the National Health Insurance Service. To validate the result, a hospital cohort of patients with Pca consisting of GnRHa users (n = 205) and nonusers (n = 479) in a tertiary referral center from January 1, 2006 to December 31, 2016, were also analyzed. Traditional and propensity score-matched Cox proportional hazards models were used to estimate the effects of ADT on the risk of dementia and Parkinson's disease. In univariable analysis, risk of dementia was associated with GnRHa use in both nationwide and hospital validation cohort (hazard ratio [HR], 1.696; 95% CI, 1.425–2.019, and HR, 1.352; 95% CI, 1.089–1.987, respectively). In a nationwide cohort, ADT was not associated with dementia in both traditional and propensity score-matched multivariable analysis, whereas in a hospital validation cohort, ADT was associated with dementia only in unmatched analysis (HR, 1.203; 95% CI, 1.021–1.859) but not in propensity score-matched analysis. ADT was not associated with Parkinson's disease in either nationwide and validation cohorts. This population-based study suggests that the association between GnRHa use as ADT and increased risk of dementia or Parkinson's disease is not clear, which was also verified in a hospital validation cohort.

from National Health Insurance Service (NHIS). Data are available from the Healthcare Bigdata Hub of Korea (contact via https://opendata.hira.or.kr/) for researchers who meet the criteria for access to confidential data.

**Funding:** This research was supported by Hallym University Research Fund 2018(H20180251) received by MS. URL: https://sanhak.hallym.ac.kr/ The funders had no role in study design, data collection and analysis, decision to publish, or preparation of the manuscript.

**Competing interests:** The authors have declared that no competing interests exist.

## Introduction

In recent decades, androgen deprivation therapy (ADT) using gonadotropin-releasing hormone agonist (GnRHa) has dramatically increased [1] because it has not only been widely used for the treatment of metastatic prostate (Pca) but also loco-regional disease with high-risk features given the fact that randomized evidence supports the use of ADT in combination with external beam radiation therapy [2, 3]. Although ADT has demonstrated delayed progression and survival benefit [4], several studies have reported various adverse effects related to ADT [5–7]. In particular, the risk of cognitive impairment has been suggested in the previous study because it is known to be associated with decreased testosterone levels and the ADT effects are based on its ability of suppressing the testosterone level [8, 9].

Recent published studies have conflicting results, making it difficult to draw conclusion on the impact of ADT on dementia and Parkinson's disease [10–13]. These discrepancies may have occurred because of important methodologic limitations and errors in statistical interpretation, especially after adjusting confounding variables [11]. Confounding may occur when the false association is created due to differences between patients that are related to both exposure (e.g., treatment assignment) and outcome; for example, patients receiving ADT tend to be older, have low socioeconomic status (SES), and have more comorbidities [11, 12] and these are also associated with increased risk of dementia. In addition, results from nationwide datasets should be validated because it does not report clinical variables, such as alcohol consumption, smoking status, and body mass index (BMI).

In this nationwide population-based cohort study, the impact of ADT using GnRHa on the risk of dementia and Parkinson's disease in patients with Pca was investigated using the traditional multivariable regression and propensity score adjustment to precisely control confounding variables, and the results were validated in a hospital cohort to estimate the effects of confounders not measured in the nationwide cohort.

## Materials and methods

### Study population: Nationwide and hospital validation cohort

To conduct this nationwide, population-level, historical cohort study on Pca patients with or without ADT, data from the historical cohort of 579,377 men with International Classification of Disease, Tenth Revision (ICD-10) code C61 for Pca was collected from the National Health Insurance Service (NHIS) database between January 1, 2012 and December 31, 2016. Because NHIS is a single-payer, universal health coverage system and also includes Medicaid program, its database consists the claim data of more than 99% of the entire population (approximately 50 million people). Accordingly, the database provides comprehensive information on the diagnosis, prescriptions, procedures, sociodemographic status, and comorbidities. NHIS database includes detailed information regarding the national insurance claims (for procedures such as surgeries and/or medical treatment) by a physician, diagnostic codes determined also by the physician, and patients' demographic parameters. Because the entire Korean population is covered by either a National Health Insurance or Medicaid (approximately 97% and 3%, respectively), the number of patients collected would include the entire Pca patients in Korean population during the study period.

Patients who were newly diagnosed with Pca within the same period (from July 1, 2012, to December 31, 2012) and have follow-up periods of at least 48 months were included. Among the patients with C61, men with 2 claim records in a row under C61 within 6 months from July 1, 2012, to December 31, 2012, were included to minimize confounding effects due to diagnostic errors, and those who had records from January 1, 2012, to June

31, 2012, were excluded; therefore, only newly diagnosed Pca patients were included. Patients receiving chemotherapy, antimuscarinics, and psychiatric drugs, previously diagnosed dementia, with neurological disease, and/or with Parkinson's disease were also excluded. Codes used to identify these diseases are shown below (S1 Table). In addition, patients who did not show a prostate biopsy billing code (C8551, C8552) was excluded from the analysis. To obtain the precise result of chemical castration effect on our diseases of interest, patients with bilateral orchiectomy were also excluded from the analysis. All patients were required to have 6 months follow-up (from January 1 to June 30, 2012) before the cohort entry to strictly follow the provided inclusion criteria. Finally, a total of 7,324 patients who were newly diagnosed with Pca during the aforementioned period and had records of follow-up visit though December 31, 2016 were included in the analysis (indicating all patients were diagnosed as Pca within the same period and have follow-up periods of at least 48 months).

GnRHa users were defined as patients who first used GnRHa from July 1, 2012, to December 31, 2012, and had at least 1-year exposure. A total of 3,201 GnRHa users and 4,123 nonusers were identified by reviewing the claim data for receipt of GnRHa throughout the study period (S2 Table). The duration of GnRHa exposure was calculated as the sum of 1-month equivalent doses.

Because NHIS database do not include detailed clinicopathologic data, such as alcohol consumption and smoking status, hospital cohort of patients with Pca with or without using GnRHa as ADT in a tertiary referral center was separately analyzed for validation of a nationwide cohort. Medical records of 1,735 consecutive patients who visited the outpatient clinic from January 1, 2006 to December 31, 2016, for the treatment of Pca was reviewed and 684 patients were finally included in the analysis after excluding those with previous chemotherapy, history of dementia, neurological disease, and/or Parkinson's disease, incomplete follow-ups, and inaccurate medical records. The institutional review board of Hallym University Sacred Heart Hospital, Republic of Korea (No. 2017-I106) approved this study and granted a waiver of informed consent from study participants because the NHIS database only provided anonymous identification code and because of its retrospective nature and the NHIS database only provided anonymous identification codes.

## Outcomes and other covariates

The primary outcome was development of dementia and Parkinson's disease. The end of the follow-up period was the time of development of the primary outcome, or December 31, 2016 (Fig 1). Prevalent dementia and Parkinson's disease were defined as either condition diagnosed between January 1, 2012, and December 31, 2012, to exclude those outcomes occurring before the Pca diagnosis. We defined incident dementia and Parkinson's disease when the condition occurred after the enrollment (after January 1, 2013). All diagnoses, including outcomes and covariates, were ascertained using the ICD-10 diagnostic codes (S1 Table). To improve diagnostic accuracy and avoid over- and underestimation of primary outcomes, we only included patients with dementia or Parkinson's disease with two and more diagnoses during hospitalization or at the outpatient clinic.

Demographic data, medication use history (S2 Table), use of antiandrogen, medical history, and Charlson comorbidity index (CCI) score were abstracted as adjustment covariates from the NHIS database. In hospital validation cohort, alcohol consumption, smoking status, BMI, PSA, biopsy Gleason score, and clinical T stage were also included as covariates by retrospectively reviewing the medical records. Alcohol consumption [14], and smoking status [15] were categorized according to previous studies. Socioeconomic status (SES) was estimated by

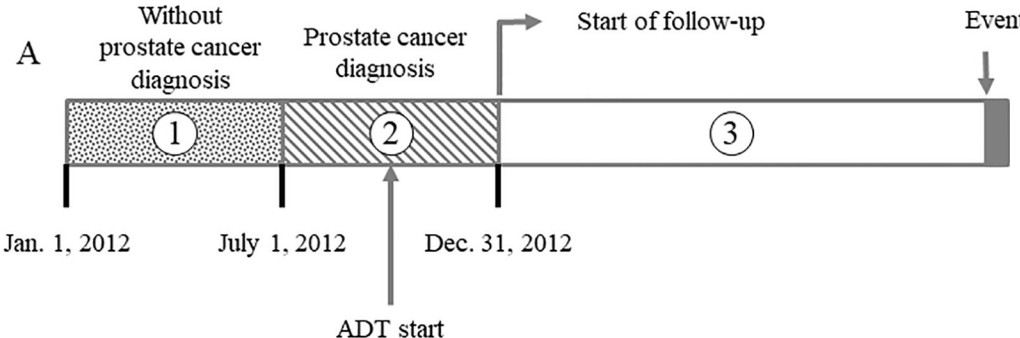

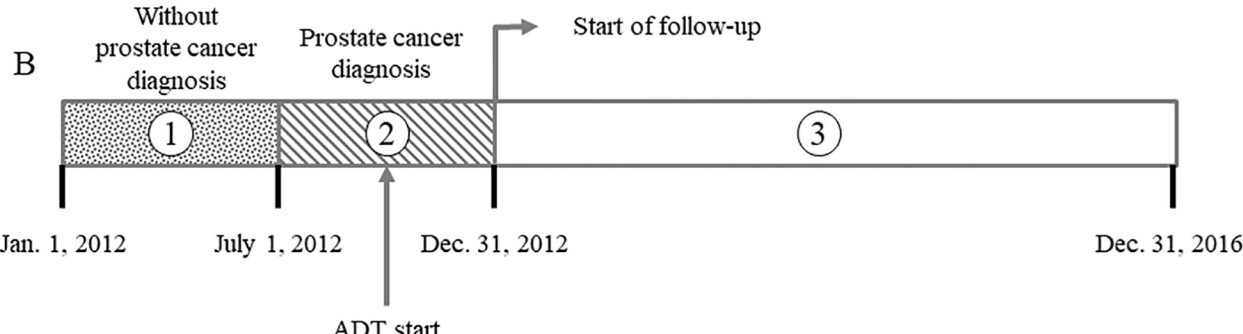

**Fig 1. Criteria for cohort entry.** Patients without prostate cancer in first 6 months (period 1) and with prostate cancer in the next 6 months (period 2) were enrolled in to the cohort only to include those who were newly diagnoses as prostate cancer. During period 2, GnRHa users were not only diagnosed with prostate cancer but also administered ADT. Patients who developed dementia or Parkinson's disease during periods 1 + 2 were defined as patients with prevalent dementia and/or Parkinson's disease and were excluded from the analysis. Patients who developed dementia or Parkinson's disease (A) may have different follow-up period as the time of development of such diseases may differ. However, patients who did not developed dementia or Parkinson's disease (B) had the exactly same follow-up period.

categorizing the patients according to the type of insurance: National Health Insurance and Medicaid.

## Statistical analysis

Patient characteristics, past medical history, and comorbidities were described based on GnRHa use. The difference of covariatesbetween GnRHa users and nonusers were compared using the $\chi^2$ test. Kaplan-Meier method was used to estimate the cumulative incidence curves for dementia and Parkinson's disease. To assess the effects of GnRHa use on dementia and Parkinson's disease, traditional unadjusted, multivariable-adjusted, and propensity score matched Cox proportional hazards models were used to calculate hazard ratios.

For propensity score matching, all covariates used in the multivariable analysis were included in matching for the nationwide cohort. Alcohol consumption, smoking status, BMI, and PSA were added as covariates in the hospital validation cohort for matching. Effects of GnRHa use duration on the outcomes were also included for analysis by defining the duration as ≤12, 13–24, 25–36, and ≥37 months. All statistical analyses were performed using SAS®-Software 9.3, (SAS Institute Inc., Cary, NC), and a *p*-value of <0.050 was considered statistically significant.

## Results and discussion

### Baseline characteristics

Among the patients in the nationwide cohort, 354 (4.8%) and 125 (1.7%) were newly diagnosed with dementia and Parkinson's disease, respectively, during the follow-up period. Before matching, GnRHa users were significantly older, reside less in the urban area, have lower SES, and higher comorbidities. However, these measured covariates showed no statistically significant differences after a propensity score matching. Table 1 presents the characteristics of the nationwide cohort, categorized by GnRHa use.

The hospital validation cohort was composed of 205 GnRHa users and 479 nonusers as control group, and 34 (4.9%) patients developed dementia and/or Parkinson's disease during the study period. GnRHa users in the validation cohort were older ($p = 0.0019$), reside less in the urban area ($p = 0.0026$), and have higher comorbidities ($p = 0.0008$) than nonusers, which is similar to those in the nationwide cohort. However, SES did not significantly differ between the two groups ($p = 0.0612$). The degree of drinking alcohol and smoking was higher in the GnRHa uses than nonusers ($p = 0.0074$, <0.0001, respectively), whereas BMI did not show a significant difference between the two groups ($p = 0.5384$). After matching, all baseline characteristics between the two groups were balanced (S3 Table).

### Clinical outcomes in the nationwide cohort

Although univariable analyses presented by the Kaplan-Meier curve showed that GnRHa use significantly increased the development of dementia (log rank, $p < 0.0001$; Fig 2A), the multivariable Cox regression analysis revealed that GnRHa use was not independently associated with dementia (hazard ratio [HR], 0.912; 95% confidence interval [CI], 0.759–1.097; $p = 0.3280$; Table 2). After adjusting for measured confounders using the multivariable and propensity score matching, GnRHa use remained not associated with dementia (HR, 0.957; 95% CI, 0.798–1.085; $p = 0.5457$; Table 3). The development of Parkinson's disease was also not associated with GnRHa uses in the univariable Kaplan-Meier analysis (log rank, $p = 0.0941$; Fig 2B) and propensity score-adjusted multivariable analysis (HR, 1.195; 95% CI, 0.958–1.518; $p = 0.2651$; Table 3). The impact of GnRHa use duration on the outcomes were also measured as shown in Table 2; however, GnRH use sustained its non-relevance to the development of both dementia and Parkinson's disease on the multivariable Cox regression models. Significant predictors of dementia and Parkinson's disease are shown in Table 2.

### Clinical outcomes in the hospital validation cohort

The results in our hospital validation cohort were not much different from those in the nationwide cohort. However, GnRHa uses were independently associated with increased risk of dementia (HR, 1.203; 95% CI, 1.021–1.859; $p = 0.0327$) in the unmatched multivariable analysis, but it lost its significance after the propensity score matching (HR, 1.075; 95% CI, 0.865–1.778; $p = 0.1365$). GnRHa uses were not associated with Parkinson's disease in all analyses, including the unadjusted, multivariable, and propensity score matched models (Table 3). The duration of GnRH uses was also not significantly associated with the outcomes in the validation cohort (Table 2). Factors that are independently associated with increased risk of dementia and Parkinson's disease are also shown in Table 2.

In this nationwide population-based cohort study, patients undergoing ADT using GnRHa did not show increased risk of dementia and Parkinson' disease, which was consistently observed in the multivariable-adjusted and propensity score-matched analyses. The relationship between the duration of the GnRHa use and the development of dementia and

**Table 1. Distribution of baseline characteristics across GnRHa users or nonusers before and after propensity score adjustment in the nationwide cohort.**

| Characteristics | Full cohort | | | Propensity score-matched cohort | | |
|---|---|---|---|---|---|---|
| | GnRHa† Users N = 3,201 | GnRHa Nonusers N = 12,123 | *p*-value | GnRHa Users N = 3,201 | GnRHa Nonusers N = 3,228 | *p*-value |
| Age at diagnosis, year (Mean ±SD) | 72.8±8.5 | 66.8±6.9 | <0.0001 | 72.8±8.5 | 72.6±7.3 | 0.4105 |
| **Residence** | | | 0.0006 | | | 0.8953 |
| Urban | 2,462 (76.9) | 9,723 (80.2) | | 2,462 (76.9) | 2,486 (77.0) | |
| Suburban/rural | 739 (23.1) | 2,400 (19.8) | | 739 (23.1) | 742 (23.0) | |
| **Insurance type** | | | 0.0002 | | | 0.2021 |
| National health insurance | 3,001 (93.8) | 11,601 (95.7) | | 3,001 (93.8) | 3,037 (94.1) | |
| Medicaid | 200 (5.3) | 522 (4.3) | | 200 (5.3) | 191 (5.9) | |
| **Prior medication use** | | | | | | |
| Statin | 392 (12.3) | 1,370 (11.3) | 0.2129 | 392 (12.3) | 399 (12.4) | 0.8590 |
| Antihypertensive | 555 (17.3) | 1,915 (15.8) | 0.0861 | 555 (17.3) | 561 (17.4) | 0.8656 |
| Anticoagulants | 345 (10.8) | 987 (8.1) | <0.0001 | 345 (10.8) | 347 (10.7) | 0.6985 |
| Antiplatelet therapy | 283 (8.8) | 867 (7.2) | 0.1740 | 283 (8.8) | 288 (8.9) | 0.7931 |
| **Prior antiandrogen use** | 510 (15.9) | 109 (0.9) | <0.0001 | 510 (15.9) | 45 (1.4) | <0.0001 |
| **Medical history** | | | | | | |
| Hypertension | 1,607 (50.2) | 5,855 (48.3) | 0.1134 | 1,607 (50.2) | 1,603 | 0.7516 |
| Diabetes mellitus | 712 (22.2) | 2,813 (23.2) | 0.3518 | 712 (22.2) | 709 (22.0) | 0.3725 |
| Hyperlipidemia | 647 (20.2) | 2,291 (18.9) | 0.0687 | 647 (20.2) | 651 (20.2) | 0.8428 |
| Cardiovascular disease | 670 (20.9) | 1,612 (13.3) | 0.0342 | 670 (20.9) | 656 (20.3) | 0.3924 |
| Liver disease | 105 (3.3) | 497 (4.1) | 0.0750 | 105 (3.3) | 98 (3.0) | 0.1194 |
| Other cancer | 408 (12.8) | 1,237 (10.2) | 0.0006 | 408 (12.8) | 393 (12.2) | 0.2598 |
| Chronic kidney disease | 129 (4.0) | 400 (3.3) | 0.0752 | 129 (4.0) | 119 (3.7) | 0.3581 |
| COPD‡ | 292 (9.1) | 1,176 (9.7) | 0.3812 | 292 (9.1) | 293 (9.1) | 0.9531 |
| Asthma | 230 (7.2) | 812 (6.7) | 0.4350 | 230 (7.2) | 241 (7.5) | 0.5543 |
| Peripheral vascular disease | 58 (11.2) | 1,309 (10.8) | 0.6186 | 58 (1.8) | 61 (1.9) | 0.7889 |
| **Charlson comorbidity index** | | | <0.0001 | | | 0.4528 |
| 0, 1 | 128 (4.0) | 1,307 (10.8) | | 128 (4.0) | 137 (4.2) | |
| 2 | 399 (12.5) | 2,559 (21.1) | | 399 (12.5) | 416 (12.9) | |
| 3 | 878 (27.4) | 3,346 (27.6) | | 878 (27.4) | 892 (27.6) | |
| 4 | 875 (27.3) | 2,552 (21.1) | | 875 (27.3) | 866 (26.8) | |
| ≥5 | 921 (28.8) | 2,359 (19.5) | | 921 (28.8) | 917 (30.1) | |
| **Other treatment** | | | | | | |
| Radical prostatectomy | 0 (0.0) | 7,492 (61.8) | <0.0001 | 0 (0.0) | 1,023 (31.7) | <0.0001 |
| Radiotherapy | 691 (21.6) | 2,291 (18.9) | 0.0265 | 691 (21.6) | 1,169 (36.2) | 0.0127 |

Values are presented as number (%) unless otherwise indicated.

†gonadotropin-releasing hormone agonist,

‡chronic obstructive pulmonary disease

Parkinson's disease was analyzed in the same manner, but no correlation was found. This finding was also validated in a hospital cohort, which showed similar results without significant association after the propensity-score matching on the application of the same robust statistical methods.

Recently, reports from several population-based observational studies had been released, showing conflicting results [10–13, 16]. These discrepancies in results may have originated

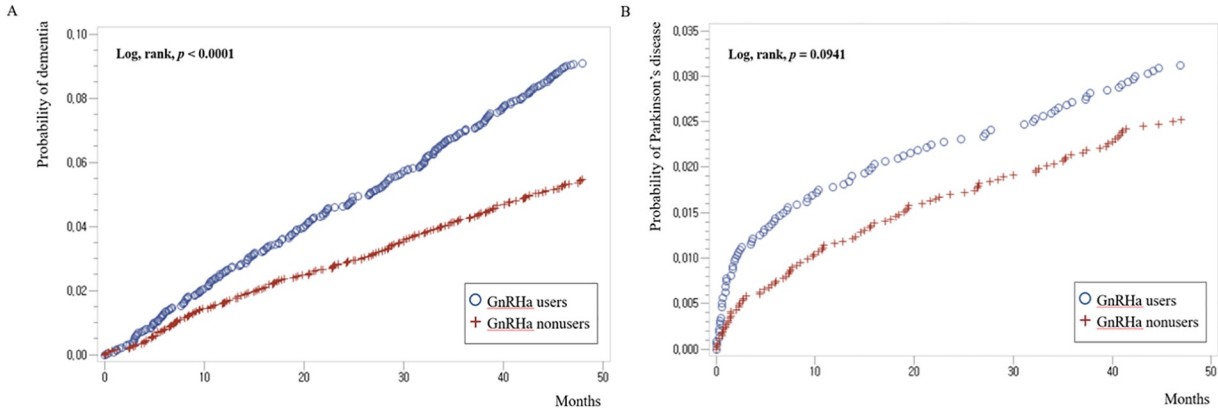

**Fig 2. Kaplan-Meier curves for the development of (A) dementia and (B) Parkinson's disease, compared according to GnRHa users and nonusers in nationwide cohort.**

from the differences in patient cohort characteristics, end-point settings (dementia, cognitive impairment, Alzheimer's disease, and etc.) and statistical techniques controlling confounders. Particularly, studies showing that ADT increases the risk of cognitive impairment or dementia had different start of follow-up between ADT users and nonuser, because the former started upon the initiation of ADT whereas the latter started at the time of diagnosis plus the median time of ADT use [13, 16]. Therefore, in these reports, the average follow-up period for ADT users may actually be longer than those for nonusers because the ADT treatment may have been initiated years after their diagnosis which leads to overestimation of dementia risks due to immortal time bias [17]. Moreover, these studies did not include adjustment for important confounder, including lifestyle variables, such as alcohol consumption and smoking status, which are known to have a strong impact on cognitive impairment and Parkinson's disease [18, 19].

Our study was specifically designed to minimize potential, unmeasured bias that previous studies might have for more accurate conclusions. First, the longitudinal population-based Korean NHIS database was used in our study to allow identification of all ADT cases using GnRHa for the treatment of Pca and to investigate the correlation between ADT and dementia in the entire population of the Republic of Korea (about 50 million) rather than the specific group of populations. Second, the strictest criteria for the cohort entry were established among the recent population-based cohort study. All patients were diagnosed at the same period of time (from July 1, 2012, to December 31, 2012) and had similar and contemporary follow-up period (from the time of diagnosis to December 31, 2016), and those with C61 code from January 1, 2012, to June 31, 2012 were excluded, so that only newly diagnosed patients are included to eliminate immortal time bias [20]. In addition, we performed GnRHa-based analysis, excluding patients receiving other forms of ADT, such as surgical orchiectomy, for better accuracy, and of course, antiandrogen was included in the analysis as a covariable. Third, the addition of analysis using hospital validation cohort allowed us to supplement estimation of important potential confounders' effects on the result, such as alcohol consumption, smoking status, and oncological characteristics, which are often missing in administrative nationwide databases. Finally, the results in this study were double checked by multiple statistical techniques such as traditional multivariable regression and propensity score adjustment, to eliminate any unmeasured bias based on differences in patient characteristics between ADT users and nonusers.

**Table 2. Multivariate Cox regression model showing hazard ratios associated with GnRHa use relative to non-use.**

| | HR† (95% CI‡) | | | |
|---|---|---|---|---|
| | Nationwide cohort | | Hospital validation cohort | |
| Characteristics | Dementia | Parkinson's disease | Dementia | Parkinson's disease |
| GnRHa§ use (Use vs Non-use) | 0.912 (0.759–1.097) | 1.495 (0.846–1.985) | **1.203 (1.021–1.859)** | 1.226 (0.847–1.554) |
| GnRHa use (Duration month) | | | | |
| No use | Reference | Reference | Reference | Reference |
| ≤12 | 0.985 (0.921–1.792) | 1.245 (0.847–1.268) | 1.238 (0.784–2.315) | 1.351 (0.784–2.213) |
| 13–24 | 1.320 (0.912–1.847) | **1.635 (1.024–2.658)** | 1.138 (0.842–1.957) | 1.284 (0.849–1.984) |
| 25–36 | 1.151 (0.879–1.314) | 1.358 (0.926–1.874) | 1.864 (0.789–2.527) | **1.405 (1.021–2.791)** |
| ≥37 | **1.246 (1.018–1.654)** | 0.968 (0.878–1.315) | 1.358 (0.889–1.715) | 0.896 (0.542–1.573) |
| Age at diagnosis (continuous) | **1.157 (1.144–1.171)** | **1.098 (1.078–1.118)** | **1.498 (1.085–1.983)** | **1.154 (1.002–1.476)** |
| Urban vs. Suburban/rural | **2.206 (1.449–3.357)** | 0.904 (0.736–1.111) | **2.195 (1.217–4.318)** | 1.109 (0.894–1.231) |
| National health insurance vs. Medicaid | 1.336 (0.992–1.801) | 1.152 (0.643–2.068) | **2.357 (1.196–5.327)** | 1.895 (0.799–3.215) |
| Alcohol consumption | | | | |
| None | | | Reference | Reference |
| Light | | | 0.982 (0.624–1.983) | 0.782 (0.239–1.057) |
| Moderate | | | 1.128 (0.895–2.854) | 0.987 (0.752–1.254) |
| Heavy | | | **1.758 (1.121–3.378)** | 1.015 (0.897–1.337) |
| Smoking status (pack years) | | | | |
| Never smokers | | | Reference | Reference |
| Ex-smokers | | | **2.021 (1.207–5.547)** | 0.987 (0.568–1.027) |
| <30 | | | 1.231 (0.589–3.249) | 1.054 (0.758–1.568) |
| 30–59 | | | **2.215 (1.059–4.389)** | 2.147 (0.989–3.847) |
| ≥60 | | | **2.846 (1.894–5.517)** | 1.548 (0.897–2.257) |
| Body mass index (kg/m²) | | | | |
| <18.5 | | | Reference | Reference |
| 18.5–22.9 | | | 0.768 (0.324–1.129) | 1.023 (0.754–1.239) |
| 20.3–24.9 | | | 1.005 (0.898–1.327) | 0.784 (0.356–1.087) |
| ≥25 | | | **0.754 (0.215–0.987)** | 0.895 (0.687–1.257) |
| Prostate-specific antigen (ng/mL) | | | | |
| <30 to ≥30 | | | 0.985 (0.855–1.224) | 0.874 (0.635–1.215) |
| Biopsy Gleason score | | | | |
| ≤6 | | | Reference | Reference |
| 7 | | | 1.089 (0.591–3.514) | 1.058 (0.512–2.068 |
| ≥8 | | | 0.957 (0.352–1.258) | 1.125 (0.869–2.872) |
| Clinical T stage | | | | |
| T1 | | | Reference | Reference |
| T2 | | | 1.268 (0.598–3.647) | 0.895 (0.354–1.851) |
| T3, T4 | | | 1.154 (0.854–2.324) | 1.129 (0.753–2.534) |
| Prior medication use | | | | |
| Statin | 1.177 (0.914–1.517) | 1.416 (0.971–2.065) | 1.298 (0.846–1.627) | 1.374 (0.842–2.316) |
| Antihypertensive | 0.859 (0.623–1.168) | 1.095 (0.895–1.421) | 1.138 (0.895–1.216) | 0.875 (0.452–2.257) |
| Anticoagulant | 1.085 (0.809–1.456) | 0.806 (0.522–1.243) | 1.321 (0.714–1.592) | 0.748 (0.428–2.428) |
| Antiplatelet therapy | 1.041 (0.782–1.387) | 0.872 (0.516–1.476) | **1.089 (1.001–1.485)** | 1.125 (0.754–1.369) |
| Prior antiandrogen use | 0.895 (0.684–1.028) | 0.986 (0.858–1.089) | 1.215 (0.842–2.287) | 0.912 (0.784–1.125) |
| Medical history | | | | |
| Hypertension | 1.050 (0.883–1.247) | 0.917 (0.697–1.208) | 1.151 (0.954–1.841) | 1.315 (0.774–2.297) |
| Cerebrovascular disease | **1.345 (1.103–1.640)** | **2.024 (1.522–2.694)** | 1.458 (0.987–2.875) | 1.248 (0.684–3.114) |

*(Continued)*

**Table 2.** (Continued)

| | HR† (95% CI‡) | | | |
|---|---|---|---|---|
| | Nationwide cohort | | Hospital validation cohort | |
| Characteristics | Dementia | Parkinson's disease | Dementia | Parkinson's disease |
| Diabetes mellitus | **1.335 (1.098–1.622)** | 1.072 (0.778–1.479) | 1.254 (0.571–2.482) | 0.987 (0.541–2.357) |
| Hyperlipidemia | | | | |
| Liver disease | 1.045 (0.669–1.634) | 0.514 (0.191–1.382) | 0.873 (0.425–2.185) | 1.011 (0.846–2.989) |
| Other cancer | 0.803 (0.597–1.081) | 0.947 (0.609–1.475) | 1.087 (0.899–2.154) | 0.867 (0.542–2.318) |
| Chronic kidney disease | 1.147 (0.741–1.775) | 0.955 (0.450–2.029) | 0.987 (0.658–1.987) | 1.021 (0.687–1.855) |
| COPD¶ | 1.116 (0.884–1.476) | 1.469 (0.980–2.203) | 1.284 (0.987–3.157) | 1.358 (0.784–3.124) |
| Asthma | 1.098 (0.796–1.514) | 0.841 (0.469–1.507) | 0.684 (0.387–1.842) | 1.021 (0.754–2.357) |
| **Charlson comorbidity index (continuous)** | **1.039 (1.018–1.091)** | 1.054 (0.949–1.038) | **1.021 (1.004–1.241)** | 1.010 (0.921–1.108) |
| **Radical prostatectomy** | 1.127 (0.821–1.935) | 1.024 (0.732–1.527) | 1.527 (0.812–2.579) | 0.998 (0.641–1.021) |
| **Radiation therapy** | 1.216 (0.715–2.054) | 1.324 (0.824–2.578) | 1.427 (0.859–2.365) | 1.268 (0.887–1.694) |

Values in bold type are statistically significant at p < 0.05

†HR hazard ratio,

‡CI confidence interval,

§GnRHa gonadotropin-releasing hormone agonist,

¶Chronic obstructive pulmonary disease

Almost all previous studies clearly showed the difference in patient characteristics between the ADT users and nonusers, that is, ADT users were older, usually heavier smoker, and had more serious comorbidities [10–13, 16, 20]. Indeed, our result similarly demonstrated differences between the two groups in several ways in both the nationwide and hospital validation cohorts. Patients are also more likely to receive definitive radiotherapy and subsequent ADT if they are unsuitable to surgery secondary to medical comorbidities, such as cardiovascular

**Table 3. Unadjusted and adjusted Cox proportional hazard regression analysis associated with GnRHa use relative to non-use.**

| Variables | Dementia | | Parkinson's | |
|---|---|---|---|---|
| | HR† (95% CI‡) | p-value | HR (95% CI) | p-value |
| **Nationwide cohort** | | | | |
| Unadjusted model | **1.696 (1.425–2.019)** | **<0.0001** | 1.245 (0.946–1.639) | 0.1174 |
| Multivariable model | 0.912 (0.759–1.097) | 0.3280 | 1.495 (0.846–1.985) | 0.1321 |
| Propensity score model | 0.957 (0.798–1.085) | 0.5457 | 1.195 (0.958–1.518) | 0.2651 |
| **Hospital validation cohort** | | | | |
| Unadjusted model | **1.352 (1.089–1.987)** | **0.0214** | 1.398 (0.851–1.987) | 0.0954 |
| Multivariable model | **1.203 (1.021–1.859)** | **0.0327** | 1.226 (0.847–1.554) | 0.3418 |
| Propensity score model | 1.075 (0.865–1.778) | 0.1365 | 1.515 (0.845–2.398) | 0.8427 |

*Note* A hazard ratio (HR) of > 1 indicates increased risk of dementia and Parkinson's disease in GnRHa users. Multivariable and propensity score models adjusted for age at diagnosis, residence (urban vs. suburban/rural), socioeconomic status (National health insurance vs. Medicaid), prior medication, prior antiandrogen use, past medical history, and comorbidity index for the nationwide cohort. For hospital validation cohort, alcohol consumption, smoking status, body mass index, and prostate-specific antigen were added for adjustment.

Values in bold are statistically significant at *p* < 0.05

†HR hazard ratio,

‡CI confidence interval,

disease, which are also considered strong risk factors for cognitive impairment [21]. Eventually, covariates showing differences between the two groups may possibly affect not only the development of dementia and Parkinson's disease but also administration of ADT, and these are likely to have led to bias in deriving conclusions. According to our results, covariables that showed significant difference between GnRHa users and nonusers, such as age, residence, SES, and CCI, were also in fact significant factors associated with risk of dementia in the nationwide or hospital validation cohort. In addition, patients receiving ADT are more likely to be diagnosed with different diseases, including dementia and Parkinson's disease, because they may usually have to visit hospitals more frequently than ADT nonusers. Furthermore, in the result of our hospital validation cohort, GnRHa use appeared to be related to the risk of dementia in the traditional multivariable analysis; however, this relationship was not found in the propensity-score matched analysis. This result suggests that when comparing the two groups, failure to find a precise adjustment on the differences between the groups may lead to incorrect conclusions. After establishing strict criteria for cohort entry and precisely adjusting potential confounders, ADT with GnRHa use was not associated with dementia and Parkinson's disease.

Nevertheless, several limitations should be considered in this study. First, misclassifications and diagnosis errors are possible, because identifying dementia and Parkinson's disease was limited to the diagnostic codes recorded not only by specialist but also general practitioner, and patients with mild symptoms of dementia might not have sought medical services. Second, although we have validated our results from nationwide cohort by adding strong risk factors for dementia such as smoking status and alcohol consumption, as well as cancer-related factors, such as Gleason score and clinical stage to hospital validation cohort, those covariables were not actually tested in the nationwide cohort. Furthermore, potential confounders may be more diverse that family history of dementia or Parkinson's disease, physical activity, dietary habits, and educational background that were not included in the hospital validation cohort may be associated with the risk of dementia and potentially caused bias. One significant issue is that the statistical modeling used in this study still may not be perfect in comparing the groups that has intrinsically different characteristics. However, authors from the United Kingdom indicated that such unmeasured or unknown confounders are modestly associated with the outcomes and are unlikely to have biased the result under most reasonable assumptions [11]. Lastly, the follow-up period was limited to 4 years due to limited access to data. It is possible that the time of 4 years does not sufficiently reflect the effect of prolonged exposure to ADT. Nevertheless, most of the studies to date also reported a mean follow-up period of between 4.3 and 5 years, which was similar to our study [11, 12, 16, 20]. More accurate results are expected if a longer period of nationwide population data is available.

## Conclusions

The outcomes of this population-based cohort study suggest that the association of GnRHa use with increased risk of dementia and Parkinson's disease is not clear enough to effect current clinical practice. Further long-term study is warranted for the clarification.

## Supporting information

**S1 Table. Codes used to identify diagnosis.**
(DOCX)

**S2 Table. Codes used to identify medications.**
(DOCX)

**S3 Table. Distribution of baseline characteristics across GnRH agonist users or nonusers before and after the propensity score adjustment in the hospital validation cohort.**
(DOCX)

**S1 Data.**
(DOCX)

## Author Contributions

**Conceptualization:** Myungsun Shim, Jin Seon Cho.

**Data curation:** Myungsun Shim, Woo Jin Bang, Cheol Young Oh, Yong Seong Lee.

**Formal analysis:** Myungsun Shim, Jin Seon Cho.

**Funding acquisition:** Myungsun Shim.

**Investigation:** Myungsun Shim.

**Methodology:** Myungsun Shim, Young-Su Ju.

**Project administration:** Myungsun Shim.

**Software:** Young-Su Ju.

**Supervision:** Seong Soo Jeon, Hanjong Ahn, Young-Su Ju, Jin Seon Cho.

**Writing – original draft:** Myungsun Shim.

**Writing – review & editing:** Myungsun Shim.

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
