## [Decision Letter · Decision Letter 0]

24 Jul 2020

PONE-D-20-17656

Risk of dementia and Parkinson's disease in patients treated with androgen deprivation therapy using gonadotropin-releasing hormone agonist for prostate cancer: A nationwide population-based cohort study.

PLOS ONE

Dear Dr. Cho,

Thank you for submitting your manuscript to PLOS ONE. After careful consideration, we feel that it has merit but does not fully meet PLOS ONE’s publication criteria as it currently stands. Therefore, we invite you to submit a revised version of the manuscript that addresses all the points raised during the review process.

We look forward to receiving your revised manuscript.

Kind regards,

Gianluigi Forloni

Academic Editor

PLOS ONE

Journal Requirements:

Reviewers' comments:

Reviewer's Responses to Questions

**Comments to the Author**

1. Is the manuscript technically sound, and do the data support the conclusions?

Reviewer #1: Partly

2. Has the statistical analysis been performed appropriately and rigorously? 

Reviewer #1: Yes

3. Have the authors made all data underlying the findings in their manuscript fully available?

Reviewer #1: Yes

4. Is the manuscript presented in an intelligible fashion and written in standard English?

Reviewer #1: Yes

5. Review Comments to the Author

Reviewer #1: In this paper the authors retrospectively analyzed a Nationwide population in Korea of all men diagnosed within a 6 month period of time with prostate cancer and whom had a minimum of 4 years of follow-up with the purpose of determining whether the use of an GnRH agonist impacted the development of dementia. It was the conclusion of the authors, that after accounting for numerous covariates, that the development of dementia is not impacted by the use of GnRH agonists but rather patients being treated with GnRH agonists have other risk factors that predispose to the development of dementia. I have several criticisms of this study. The most pertinent of those are listed below and I think would need to be addressed prior to publication.

1. This study only included patients with new diagnosis of prostate cancer. In addition, patients had to have had a prostate biopsy performed which is frequently not done for patients with widespread metastatic disease and very high PSAs. Therefore, both patients with recurrent disease and patients who would be likely be exposed to lifelong ADT (ie the highest cumulative exposure) were not included in this study.

2. The authors do highlight that the risk of incident dementia is significantly higher in the population cohort exposed to GnRH (4.8% vs 1.7%) although that fact is surprisingly absent in the abstract. The authors simply state that “cohort, ADT was not associated with dementia in both traditional and propensity score-matched multivariable analysis.” However, it should be noted that there are significant baseline differences between these two groups. What the authors are essentially implying is that there is definitely a difference in dementia rates between these two groups but they believe that their modeling has accounted for all the differences in the baseline characteristics which actually are responsible for the increase in dementia rates between the two groups (and not the actual ADT). One significant issue with that, is when groups are this intrinsically different, can you actually “model away” all those differences.

3. Further to the above point, the hospital validation cohort in their multivariate cox regression model did show that GnRH use was associated with increasing risk of dementia. However, their propensity score model did not. So in summary, there are two univariate models, both of which show an association with dementia and GnRH use. There are two multivariate models. One shows there is an association and one does not. Then there are two propensity scored models, both of which show no association. The firm conclusions of the authors is that the covariates are responsible for the different in dementia rates not the GnRH use. These were very different groups of people and you had one multivariate analysis that still showed they are linked. The authors may be too firm in their conclusions. In addition, the authors state “Important covariates have not only influenced the risk of such diseases, but also the decision of using ADT to treat Pca.” This is a hypothesis not a conclusion (albeit, likely a reasonable hypothsis). The authors provided no analysis as to the decision-making process on use of ADT and how it was affected by covariates.

4. Prospective studies have indicated that cognitive impairment with ADT may be demonstrated as early as 3 months after initiation of ADT. The authors excluded anyone with an incident diagnosis of dementia until 6 months after the study window opened in order to avoid capturing pre-existing dementia patients. These methods eliminate the possibility of including patients with abrupt cognitive changes.

5. This is a retrospective analysis in which the authors conclude there is no association between GnRH use and dementia. I would like to see the authors comments on why prospective cohort studies using validated neurocognitive instruments can detect a difference in cognition with ADT. This study would seem to suggest that the association between GnRH use and dementia is simply an artifact due to confounding. Previously conducted small prospective studies with validated instruments would seem to refute that claim.

6. PLOS authors have the option to publish the peer review history of their article (what does this mean?). If published, this will include your full peer review and any attached files.

Reviewer #1: No

---

## [Author Response · Author response to Decision Letter 0]

14 Aug 2020

Responses to review comments

We are grateful to the reviewer for their thorough review and constructive comments. We agree with all of the comments made and have revised our paper in light of these useful suggestions. Point-by-point response to each comment and suggestion is listed below.

Reviewer #1

Comment #1. This study only included patients with new diagnosis of prostate cancer. In addition, patients had to have had a prostate biopsy performed which is frequently not done for patients with widespread metastatic disease and very high PSAs. Therefore, both patients with recurrent disease and patients who would be likely be exposed to lifelong ADT (ie the highest cumulative exposure) were not included in this study.

Response: We completely agree with the reviewer’s comment that our study design poses a risk of overlooking the prolonged effect of ADT. However, we intended to target only patients newly diagnosed with prostate cancer in order to confirm the pure effect of ADT with complete control of other covariates. If the patients who have been previously diagnosed with prostate cancer and the newly diagnosed patients are mixed in the cohort, it may not be possible to rule out the possibility of dementia due to the effects of prostate cancer itself. In addition, we think that including newly diagnosed prostate cancer patients only in the analysis may have further minimized the risk of immortal time bias (Suissa S American journal of epidemiology. 2008;167(4):492-9.).

However, again, we do agree with the reviewer’s point, so that we have included the following in the Results and discussion section, 273th row as limitation of the study.

Lastly, the follow-up period was limited to 4 years due to limited access to data. It is possible that the time of 4 years does not sufficiently reflect the effect of prolonged exposure to ADT. // More accurate results are expected if a longer period of nationwide population data is available.

Comment #2. The authors do highlight that the risk of incident dementia is significantly higher in the population cohort exposed to GnRH (4.8% vs 1.7%) although that fact is surprisingly absent in the abstract. The authors simply state that “cohort, ADT was not associated with dementia in both traditional and propensity score-matched multivariable analysis.” However, it should be noted that there are significant baseline differences between these two groups. 

We understand the reviewer’s concern that there are significant baseline differences between the two groups that may hinder the appropriate comparison of the two groups. However, the number 4.8% vs 1,7% was not for the difference in incidence of dementia, but for the number of patients diagnosed with dementia and Parkinson’s disease respectively, in nationwide cohort.

Following the reviewer’s comment, we have added the below sentence in the abstract, 32th row for the clarification.

In univariable analysis, risk of dementia was associated with GnRHa use in both nationwide and hospital validation cohort (hazard ratio [HR], 1.696; 95% CI, 1.425–2.019, and HR, 1.352; 95% CI, 1.089–1.987, respectively). 

What the authors are essentially implying is that there is definitely a difference in dementia rates between these two groups but they believe that their modeling has accounted for all the differences in the baseline characteristics which actually are responsible for the increase in dementia rates between the two groups (and not the actual ADT). One significant issue with that, is when groups are this intrinsically different, can you actually “model away” all those differences.

Again, we do agree with the reviewer’s concern in techniques for modeling the two groups. Therefore, we have included the following sentence in the Results and discussion section, 270th row as a limitation of the study.

One significant issue is that the statistical modeling used in this study still may not be perfect in comparing the groups that has intrinsically different characteristics.

Comment #3. Further to the above point, the hospital validation cohort in their multivariate cox regression model did show that GnRH use was associated with increasing risk of dementia. However, their propensity score model did not. So in summary, there are two univariate models, both of which show an association with dementia and GnRH use. There are two multivariate models. One shows there is an association and one does not. Then there are two propensity scored models, both of which show no association. The firm conclusions of the authors is that the covariates are responsible for the different in dementia rates not the GnRH use. These were very different groups of people and you had one multivariate analysis that still showed they are linked. The authors may be too firm in their conclusions. 

Response: Considering the reviewer’s comment, we agree that our conclusion is too decisive although the results were different between the different models of regression analysis. 

As following the reviewer’s comment, we had completely revised the Conclusion section as shown below in 280th row.

The outcomes of this population-based cohort study suggest that the association of GnRHa use with increased risk of dementia and Parkinson’ disease is not clear enough to effect current clinical practice. Further long-term study is warranted for the clarification. 

In addition, the Abstract section, 38th row was also modified as follows.

This population-based study suggests that the association between GnRHa use as ADT and increased risk of dementia or Parkinson’s disease is not clear, which was also verified in a hospital validation cohort. 

In addition, the authors state “Important covariates have not only influenced the risk of such diseases, but also the decision of using ADT to treat Pca.” This is a hypothesis not a conclusion (albeit, likely a reasonable hypothsis). The authors provided no analysis as to the decision-making process on use of ADT and how it was affected by covariates.

Following the reviewer’s comment, we have deleted the corresponding part.

Comment #4. Prospective studies have indicated that cognitive impairment with ADT may be demonstrated as early as 3 months after initiation of ADT. The authors excluded anyone with an incident diagnosis of dementia until 6 months after the study window opened in order to avoid capturing pre-existing dementia patients. These methods eliminate the possibility of including patients with abrupt cognitive changes.

Response: We agree with the reviewer’s comment that there may be some cases with abrupt onset of cognitive impairment after ADT initiation. In our study, as shown in figure 1, ADT was initiated at any time, not at the start of the period 2. Therefore, patient should have some diversities in the exact timing of ADT initiation, which means that not all patients had 6 months period of “no-diagnosis of dementia” before ADT start. Following the reviewer’s comment, we reviewed our data again, and noticed only 3 patients who developed dementia during the period 2. 

Following sentence was mentioned in M&M section 82th row to note the exclusion criteria. 

Patients receiving chemotherapy, antimuscarinics, and psychiatric drugs, previously diagnosed dementia, with neurological disease, and/or with Parkinson’s disease were also excluded.

Comment #5. This is a retrospective analysis in which the authors conclude there is no association between GnRH use and dementia. I would like to see the authors comments on why prospective cohort studies using validated neurocognitive instruments can detect a difference in cognition with ADT. This study would seem to suggest that the association between GnRH use and dementia is simply an artifact due to confounding. Previously conducted small prospective studies with validated instruments would seem to refute that claim.

Response: As the reviewer pointed out, there are a number of previous reports on the positive association of ADT with cognitive dysfunction. Especially in 2015, a cohort study from United States reporting an association between increased risk of Alzheimer’s disease and the use of ADT was published in J Clin Oncol [Nead K, J Clin Oncol, 2016 Feb 20;34(6):566-71]. However, two years later, a similar study from United Kingdom with complete opposite results (reporting that ADT was not associated with an increased risk of dementia) was also published in the very same journal [Khosrow-Khavar F, J Clin Oncol. 2017 Jan 10;35(2):201-207]. Therefore, in this field, an accurate conclusion has not yet been reached, and controversy continues.

There may be several reasons for the conflicting results from the previous studies. Not only differences in statistical methods, or an artifact due to confounding, but also differences in patient cohort and end-point settings (dementia, cognitive dysfunction, Alzheimer’s disease, etc.) may have resulted in such a variety of results. 

Furthermore, in most studies reporting positive associations between ADT and dementia, the start of follow-up was different between users and nonusers of ADT, which may have introduced immortal time bias [Suissa S, Am J Epidemiol 167:492-499, 2008]. Specifically, this suggests that the nonusers may have had a longer average follow-up than ADT users, thereby leading to an overestimation of the hazard ratio. We believe that this immortal time bias is the most important difference between our findings and previous studies showing the association between ADT and dementia. 

Following the reviewer’s comment, we have added the following sentence in 217th row to mention the suggested reasons for different results from studies to studies.

These discrepancies in results may have originated from the differences in patient cohort characteristics, end-point settings (dementia, cognitive impairment, Alzheimer’s disease, and etc.) and statistical techniques controlling confounders.

Comments on “immortal time bias” were mentioned in the manuscript 219th row as shown below.

Studies showing that ADT increases the risk of cognitive impairment or dementia had different start of follow-up between ADT users and nonuser, because the former started upon the initiation of ADT whereas the latter started at the time of diagnosis plus the median time of ADT use.[13, 16] Therefore, in these reports, the average follow-up period for ADT users may actually be longer than those for nonusers because the ADT treatment may have been initiated years after their diagnosis which leads to overestimation of dementia risks due to immortal time bias.

---

## [Editor Report · Decision Letter 1]

15 Dec 2020

Risk of dementia and Parkinson's disease in patients treated with androgen deprivation therapy using gonadotropin-releasing hormone agonist for prostate cancer: A nationwide population-based cohort study.

PONE-D-20-17656R1

Dear Dr. Cho,

We’re pleased to inform you that your manuscript has been judged scientifically suitable for publication and will be formally accepted for publication once it meets all outstanding technical requirements.

Kind regards,

Gianluigi Forloni

Academic Editor

PLOS ONE
---

## [Editor Report · Acceptance letter]

17 Dec 2020

PONE-D-20-17656R1 

Risk of dementia and Parkinson’s disease in patients treated with androgen deprivation therapy using gonadotropin-releasing hormone agonist for prostate cancer: A nationwide population-based cohort study 

Dear Dr. Cho:

I'm pleased to inform you that your manuscript has been deemed suitable for publication in PLOS ONE. Congratulations! Your manuscript is now with our production department. 

Kind regards, 

on behalf of

Dr. Gianluigi Forloni 

Academic Editor

PLOS ONE